# Study on the Coupling Coordination between New-Type Urbanization and Water Ecological Environment and Its Driving Factors: Evidence from Jiangxi Province, China

**DOI:** 10.3390/ijerph19169998

**Published:** 2022-08-13

**Authors:** Daxue Kan, Xinya Ye, Lianju Lyu, Weichiao Huang

**Affiliations:** 1School of Economics and Trade, Nanchang Institute of Technology, Nanchang 330099, China; 2Business Administration College, Nanchang Institute of Technology, Nanchang 330099, China; 3Department of Economics, Western Michigan University, Kalamazoo, MI 49008-5330, USA

**Keywords:** new-type urbanization, water ecological environment, entropy method, coupling coordination degree model, Tobit model

## Abstract

With the rapid development of urbanization, problems such as the degradation of water ecological environment have emerged. How to improve the water ecological environment in the process of urbanization has become one of the urgent problems facing policy makers. This paper studies the coupling coordination relationship between new-type urbanization and water ecological environment, with the purpose of using insights gained from the study to help improve the quality of water ecological environment and promote sustainable development of water ecological environment. We take 11 cities in China’s Jiangxi Province as the research object, and construct the coupling coordination evaluation indicator system of new-type urbanization and water ecological environment, then using the coupling coordination degree model to examine the state of coupling coordination between new-type urbanization and water ecological environment from 2009 to 2019. We further explore its driving factors employing random effect panel Tobit model. The results show that: (1) The level of new-type urbanization in Jiangxi Province shows a steady upward trend, and the water ecological environment level tends to rise steadily and slowly, although the comprehensive score of water ecological environment in most cities is lower than 0.1, indicating that the situation of water ecological environment is not optimal yet and there is room for improvement. (2) In 2009, 2014 and 2019, the coupling coordination development level between new-type urbanization and water ecological environment in Jiangxi Province showed an upward trend, from moderate maladjustment recession to mild maladjustment recession, and from low coupling coordination to moderate coupling coordination, although the overall coupling coordination degree was low. (3) The investment in scientific and technological innovation, degree of opening-up and government capacity are positively correlated with the coupling coordination degree, while economic development level, resource agglomeration ability, education level and industrialization level are negatively correlated with the coupling coordination degree. These results can provide insights to support new-type urbanization and water ecological environment in the future, and hold great significance for urban sustainable development.

## 1. Introduction

Urbanization entails continuous concentration of population into urban areas. At present, China’s urbanization is in the stage of rapid development. The urbanization rate of permanent residents has increased from 17.92% in the early stage of reform and opening-up to 64.72% in 2021. However, with the rapid development of urbanization, problems such as the degradation of water ecological environment have emerged. The urban sewage discharge in Jiangxi Province increased from 691.13 million m^3^ in 2009 to 1036.62 million m^3^ in 2019, a 1.50 times increase. How to improve the water ecological environment level in the process of urbanization has become one of the urgent problems facing policymakers. At present, China is implementing a new-type urbanization strategy. The requirement of new-type urbanization is to continuously improve the quality of urbanization construction. Compared with the traditional formulation, the new-type urbanization emphasizes the overall improvement of internal quality, that is, to promote the transformation of urbanization from focusing on increasing the quantity and scale to focusing on improving the quality. It is a four-in-one urbanization of population, economy, society and space. In this context, this paper studies the coupling coordination relationship between new-type urbanization and water ecological environment from 2009 to 2019, with the purpose of using insights gained from the study to help improve the quality of water ecological environment, promote the sustainable development of water ecological environment, and contribute to the harmonious coexistence between man and nature.

Scholars have conducted studies focusing on the relationship between new-type urbanization and ecological environment. Firstly, in terms of theoretical research, the famous Utopian Socialist Robert Owen [1] put forward the “social environment determinism” and coordinated the relationship between ecological environment and urbanization from the perspective of ecological environment governance. Ebenezer [2] put forward the theory of “garden cities” in his book Garden Cities of Tomorrow, trying to integrate the advantages of the city with beautiful environment of the countryside through scientific planning, so as to construct a perfect ideal city. Grossman et al. [3] put forward the famous Environmental Kuznets Curve (EKC) theory and OECD [4] put forward the decoupling theory. The research on the relationship between urbanization and ecological environment in China started relatively late compared with studies on other countries, and basically began after the concept of sustainable development was put forward in the 1980s. For example, Ma and Wang [5] put forward the theory of “society-economy-nature” composite ecosystem, which laid a theoretical foundation for the research on urban ecological environment in China. Huang and Fang [6] and Fang and Yang [7] put forward the double exponential curve of interactive coupling between urbanization and ecological environment, as well as the laws on coupling system such as coupling fission law, dynamic hierarchy law, random fluctuation law, nonlinear synergy law, threshold law and early warning law. Liu et al. [8] put forward the concept, basic connotation and evolution law of “coupling magic cube” between urbanization and ecological environment, and built a theoretical analysis framework.

Secondly, in terms of empirical research, numerous scholars believe that urbanization process has a negative impact on the overall quality of ecological environment, as urbanization not only intensifies the emission of environmental pollution, but also inhibits the environmental absorption capacity [9,10,11,12,13,14,15]. Some other scholars have found that urbanization can improve ecological environment of the region, but has a certain negative impact on ecological environment quality of adjacent cities [16,17,18]. Al-Mulali et al. [19], Irfan and Shaw [20] and Li [21] studied the relationship between urbanization and ecological environment based on panel data, and obtained the evidence of an inverted U-shaped and anti-N-shaped relationship between urbanization and ecological environment, respectively. Shi [22], Zhao et al. [23], Xie et al. [24], Yao et al. [25] and Wang et al. [26] used comprehensive indicator evaluation model, random effect panel Tobit model, improved STIRPAT model, coupling coordination degree model and other research methods to empirically analyze the spatial-temporal pattern and influencing factors of the coupling coordination between new-type urbanization and ecological environment. Yang et al. [27], Liu et al. [28], Shi [29], Liu and Zhi [30], Han et al. [31], Cui et al. [32], Shang and Jiang [33], Feng et al. [34], Zhu et al. [35], Yang et al. [36], Ma et al. [37] take, respectively, Shaanxi Province, Fujian Province, China’s coastal zone, Beijing city, the Yangtze River economic belt, the Yellow River Basin, the Tibet Autonomous Region, the pan third pole region, the Wanjiang demonstration area, Cheng-Yu urban agglomeration and Nanjing city as case-study examples to calculate, analyze and evaluate the coupling coordination degree of urbanization and ecological environment through the construction of indicator system of urbanization and ecological environment evaluation. These studies revealed that due to regional differences, different selection of evaluation indicators, different local policies and economic development and other factors, the urbanization process, ecological environment and the coupling coordination between the two in each regions exhibit different characteristics.

In sum, it is found that non-Chinese scholars’ research on the relationship between new-type urbanization and ecological environment mostly focuses on theoretical aspects, while Chinese scholars mainly focus on empirical research. However, there has not been much research on the coupling coordination relationship between new-type urbanization and specifically water ecological environment, and most of the existing literatures only explore the coupling coordination relationship between new-type urbanization and ecological environment, with little further discussion on its driving factors. To fill this gap in the literature, this paper will build the evaluation indicator system of new-type urbanization and water ecological environment in Jiangxi Province, and use the entropy method, coupling coordination degree model and random effect panel Tobit model to study the coupling coordination development status between new-type urbanization and water ecological environment, and also its driving factors in Jiangxi Province and its 11 cities, in order to explore a mutually beneficial win-win road to promote the coordinated development of new-type urbanization and water ecological environment in Jiangxi Province. Compared with the research of Zhu et al. [35], Yang et al. [36], and Ma et al. [37], we further use the linear weighting method to measure the comprehensive score of new-type urbanization and water ecological environment system in Jiangxi Province based on the entropy method. We also build a new coupling coordination degree model which can fully and truly reflect the coordinated development degree between the two subsystems of new-type urbanization and water ecological environment. Furthermore, we constructs a new random effect panel Tobit model to empirically analyze the driving factors underlying the coupling coordination degree between new-type urbanization and water ecological environment to effectively tackle the result deviation caused by least square regression.

The remainder of the paper is structured as follows. Section 2 discusses research area and methods. Section 3 presents comprehensive evaluation of new-type urbanization and water ecological environment, the analysis of coupling coordination degree between new-type urbanization and water ecological environment, and analysis of the driving factors behind coupling coordination degree between new-type urbanization and water ecological environment. Finally, Section 4 summarizes the conclusions.

## 2. Research Design and Methods

### 2.1. Research Area

Jiangxi is located in Southeast China, in the middle and lower reaches of the Yangtze River, in the middle subtropical zone, with a significant monsoon climate and distinct four seasons. The province covers an area of 166,900 km^2^, with a total population of more than 45.18 million, and is divided into 11 cities (Jingdezhen, Jiujiang, Shangrao, Nanchang, Yingtan, Yichun, Xinyu, Fuzhou, Pingxiang, Ji’an, and Ganzhou), as shown in Figure 1. Jiangxi Province is endowed with rich water resources and dense river networks. The total length of rivers is about 18,400 km. It has the largest freshwater lake in China—Poyang Lake.

As one of the fastest growing urbanization areas in China, Jiangxi has achieved a relatively high socio-economic development level, with the urbanization rate reaching about 64.72% in 2021. This indicates that the development of urbanization in Jiangxi has entered a new era. However, along with continuous urbanization, the deterioration of water ecological environment in Jiangxi is becoming more and more serious, hence it is urgent for the province to improve its water ecological environment.

### 2.2. Construction of Indicator System

Based on previous studies, following the principles of comprehensiveness, scientificity and objectivity of the indicator system, and considering the availability of data, this paper constructs evaluation indicator system of new-type urbanization and water ecological environment in Jiangxi Province. Firstly, according to the definition of new-type urbanization, based on the research of Jiang et al. [38], a new-type urbanization evaluation system is constructed from four urbanization aspects: population urbanization, economic urbanization, social urbanization and spatial urbanization. We select the following specific secondary indicators to represent population urbanization: the number of urban registered unemployed, urbanization rate of permanent residents, proportion of employees in the secondary industry, and proportion of employees in the tertiary industry. The indicators of per capita GDP, growth rate of regional GDP, and proportion of tertiary industry in regional GDP are employed to evaluate economic urbanization. The extent of social urbanization is quite wide; therefore, we choose the following seven indicators to represent it: total retail sales of social consumer goods, number of museums, number of internet broadband access users, number of hospital beds, urban gas popularity rate, science and technology expenditure of the whole city, and education expenditure of the whole city. The following indicators are selected to represent spatial urbanization: proportion of urban construction land in urban area, park green area, built up area, urban road area per capita, green coverage rate of built-up area, and green space rate of built-up area. Thus, a total of 20 representative indicators are formed to measure urbanization. Secondly, we determine the indicator system of the water ecological environment through 3 primary indicators (pressure, status and response) and 16 secondary representative indicators (i.e., sewage discharge, water user population and domestic water use of urban residents, etc.) based on the current status of water resources and water environment in Jiangxi Province. The indicator systems for new-type urbanization and water ecological environment are shown in Table 1. The original data of each indicator during the period 2009–2019 were collected from China Urban Statistical Yearbook and Jiangxi Statistical Yearbook, Jiangxi Water Resource Bulletin, and Jiangxi National Economic and Social Development Statistical Bulletin.

### 2.3. Research Methodology

#### 2.3.1. Evaluation Method

The setting of weights can effectively measure and evaluate the importance of each indicator. At present, the commonly used weighting methods in the academic community mainly include objective weighting methods represented by principal component analysis, entropy weight method and factor analysis, as well as subjective weighting methods such as AHP and Delphi method. Each method has its own advantages and disadvantages. In order to avoid the one-sidedness of individual subjective weighting, this paper will adopt the entropy method of objective weighting. The specific calculation steps are as follows:

(1) The range transformation method is used to normalize the original data to eliminate the influence of different dimensions and orders of magnitude.
(1)$$\mathrm{Forward}\;\mathrm{indicator}:x_{ij}^{\prime }=\left(x_{ij}-\min x_{ij}\right)/\left(\max x_{ij}-\min x_{ij}\right)+0.0001$$
(2)$$\mathrm{Reverse}\;\mathrm{indicator}:x_{ij}^{\prime }=\left(\max x_{ij}-x_{ij}\right)/\left(\max x_{ij}-\min x_{ij}\right)+0.0001$$
where *x_ij_* is the original indicator value; max *x_ij_* and min *x_ij_* respectively represent the maximum and minimum values of the original indicators. In order to eliminate the possible abnormal negative and zero values, 0.0001 is added to the respective equation for coordinate translation. After standardization, all indicator values are between [0, 1].

(2) Calculate the proportion *Q_ij_* of the *j* indicator in region *i*.
(3)$$Q_{ij}=\frac{x_{ij}^{\prime }}{\sum_{i=1}^{n}x_{ij}^{\prime }}$$

(3) Calculate the entropy *e_j_* and coefficient of variation *g_j_* of indicator *j*.
(4)$$e_{j}=-\frac{1}{\ln m}\sum_{i=1}^{n}\left(Q_{ij}*\ln Q_{ij}\right)$$
(5)$$g_{j}=1-e_{j}\;$$

(4) Determine the indicator weight *w_j_*
(6)$$w_{j}=\frac{g_{j}}{\sum_{j=1}^{m}g_{j}}$$

#### 2.3.2. Coupling Model and Coupling Coordination Degree Model

Coupling is a concept in physics, which refers to the phenomenon that two (or more) systems interact with each other under the multiple actions of the internal system and other external systems. The degree of coupling is a measure of the degree of correlation between modules, which is used to describe the degree of influence between systems or elements [39]. Based on this, the paper will build a new coupling degree model between urbanization and water ecological environment, and the formula is as follows:(7)$$C=2\sqrt{U\times E}/(U+E)$$
where *C* refers to the coupling degree between new-type urbanization and water ecological environment, *C* ∈ [0,1], *U* refers to the new-type urbanization level, and *E* refers to the water ecological environment level. The larger the *C*, the stronger the correlation between the two subsystems, and the better the coupling effect. Conversely, if the interaction between them is weak then the coupling effect is poor. This paper mainly measures the coupling coordination between the two subsystems of new-type urbanization and water ecological environment, hence the adjustment coefficient is assigned the value of 2 as being done by some scholars.

The coupling degree can only indicate the degree of correlation between the systems, yet still cannot fully and truly reflect the degree of coordinated development between the two subsystems of new-type urbanization and water ecological environment. Therefore, the coupling coordination degree model is introduced, and the formula is as follows:(8)T=α×U+β×E
(9)$$D=\sqrt{C\times T}\;$$
where *T* is the coordination degree indicator between new-type urbanization and water ecological environment, and *D* is the coupling coordination degree; α, β are the undetermined weight coefficients. It is generally considered that the new-type urbanization and water ecological environment exert equally important impact on the coupling coordination degree of the whole system, hence both are assigned the value of 0.5 as being done by some scholars. With reference to the research results of Zhang et al. [40], it is classified according to the coupling coordination degree, as shown in Table 2.

#### 2.3.3. Model Construction of Driving Factors

Many factors affect the coupling coordination development of new-type urbanization and water ecological environment. Based on the existing relevant research results [41], and in combination with the actual situation, this paper conducts quantitative investigation by setting the degree of coupling coordination of new-type urbanization and water ecological environment as the dependent variable, and specifying the following independent variables: economic development level, resource agglomeration ability, investment in scientific and technological innovation, education level, opening-up degree, government capacity and industrialization level. See Table 3 for the explanation of specific dependent and independent variables.

In order to effectively alleviate the estimation bias caused by least square regression, this paper establishes a random effect panel Tobit model based on the panel data of 11 cities in Jiangxi Province from 2009 to 2019. The model is as follows:(10)$$D_{it}=cons+\beta_{1}eco_{it}+\beta_{2}res_{it}+\beta_{3}tech_{it}+\beta_{4}edu_{it}+\beta_{5}open_{it}+\beta_{6}gov_{it}+\beta_{7}indu_{it}+\varepsilon_{it}$$
where *D_it_* is the coupling coordination degree; *i* is the city; *t* is the year; *cons* is a constant term; *eco* is the economic development level; *res* indicates resource agglomeration ability; *tech* is the investment of scientific and technological innovation; *edu* is educational level; *open* is opening-up degree; *gov* is government capacity; *indu* is the industrialization level; *ε* is a random perturbation term. The original data of each variable were collected from China Urban Statistical Yearbook and Jiangxi Statistical Yearbook.

## 3. Research Results and Discussion

### 3.1. Evaluation Results and Discussion

By standardizing the data, the entropy method is used to determine the weights of new-type urbanization and water ecological environment assessment indicators in Jiangxi Province. The results of the respective weights are shown in Table 1. The comprehensive development indicator of new-type urbanization and water ecological environment system in Jiangxi Province and its 11 cities is further calculated by using the linear weighting method. The formula is as follows:(11)$$Y_{i}=\sum_{j=1}^{m}w_{j}\times Q_{ij}$$
where *Y_i_* is the comprehensive evaluation level of the sample *i*, *j* is the indicator type, and *m* is the number of indicators of the subsystem. Using the above formula, we obtained the comprehensive score of new-type urbanization and water ecological environment system in Jiangxi Province and its 11 cities, as shown in Figure 2 and Figure 3.

Figure 2 shows that in general, the level of new-type urbanization in Jiangxi Province exhibited a steady upward trend from 2009 to 2019. The level of new-type urbanization increased from 0.049 in 2009 to 0.14 in 2019. The upward trend became more obvious especially after 2016. This is mainly because 2016 is the first year of the 13th five-year plan, and also a crucial year for promoting supply-side structural reform. Jiangxi Province has actively responded to national policies, increased investment in science and technology, and constantly improved public facilities. Jiangxi has accelerated the transformation and upgrading of industrial structure, comprehensively deepened reform and innovation-driven development, stimulated market power and vitality, promoted social consumption, and stimulated economic development. At the same time, Jiangxi has firmly established a new development concept of innovation, coordination, green, development and sharing, increased the proportion of urban construction land, expanded the built-up areas, and promoted more coordinated development of urban and rural areas.

At the city level, Nanchang, Jingdezhen, Shangrao, Ganzhou and Yichun showed a fluctuating upward trend in the new-type urbanization level from 2009 to 2019, of which Yichun and Ganzhou showed a more prominent upward trend. The new-type urbanization level in Pingxiang city and Xinyu city continued to rise from 2009 to 2017, and after that the new-type urbanization level in Pingxiang city first rose sharply, then fell sharply, while the new-type urbanization level in Xinyu city showed a downward trend from 2017 to 2019. The new-type urbanization level in Jiujiang and Yingtan showed a fluctuating upward trend from 2009 to 2016, and an “S” shaped development pattern from 2016 to 2019. The new-type urbanization level in Ji’an city tends to be stable from 2009 to 2016, peaking from 2016 to 2017, and decreasing significantly from 2017 to 2019. The new-type urbanization level in Fuzhou city dropped sharply from 2009 to 2010, but fluctuated and increased from 2010 to 2019. Although the new urbanization level of all cities in Jiangxi Province has experienced fluctuations to a certain extent, it shows a good rising trend of development overall, indicating a bright prospect of future development of new-type urbanization in Jiangxi Province.

Figure 3 shows that the water ecological environment level in Jiangxi Province appears to rise steadily and slowly from 2009 to 2019. This slow rise could be due to several factors. (1) Due to extensive development mode of some mining areas in Jiangxi Province, the water ecological environment in mining areas is not repaired effectively, resulting in serious water and soil losses. There are also some problems in the construction of green mines, such as fraud and inadequate supervision; (2) Due to lax pollution control in the industrial park, there are serious problems of enterprises’ illegal discharge and leakage discharge, and a large number of heavy metals polluting the surrounding water ecological environment, causing great damage to the water ecological environment; (3) Due to insufficient investment in water ecological environment protection, the construction of environmental infrastructure is imperfect, the construction of sewage discharge pipe network is backward, and the urban sewage is not effectively collected, resulting in a large amount of sewage directly discharged into the river and causing serious water pollution, especially the poor water quality in the urban river section. In 2013, the Ministry of Water Resources of China launched the pilot work of building nation-wide water ecological civilization cities. Nanchang, Xinyu and Pingxiang, were selected as the first and second batch of pilot cities of national water ecological civilization, and actively improved relevant mechanisms and promoted various water ecological environment protection work on time. While the policy initiative has made positive progresses and achievements in the protection and preservation of water ecological environment in Jiangxi Province, there are still some outstanding water ecological environment problems to be addressed urgently. Jiangxi provincial government has vowed to continue to strengthen the protection of water ecological environment.

At the city level, the fluctuation of water ecological environment level in Nanchang and Jingdezhen decreased from 2009 to 2016, and rose slowly in Nanchang from 2016 to 2019, while rose prominently in Jingdezhen. In Jiujiang city, the fluctuation of the comprehensive indicator of water ecological environment decreased from 2009 to 2018, and increased significantly from 2018 to 2019, reaching the highest point. The water ecological environment level in Xinyu city showed an upward trend from 2009 to 2010, and the fluctuation decreased from 2010 to 2019. The water ecological environment level in Yingtan city first rose sharply, reaching the second peak, then fell sharply from 2009 to 2011, and then rose from 2011 to 2019. The water ecological environment level in Pingxiang city showed a downward trend from 2009 to 2011, and the fluctuation increased from 2011 to 2019. The water ecological environment level of Ganzhou, Ji’an, Yichun, Fuzhou and Shangrao cities tends to develop steadily, fluctuating around 0.1. Although all cities have experienced certain fluctuations, the comprehensive score of water ecological environment of most cities is lower than 0.1, indicating that the water ecological environment of Jiangxi Province is not yet optimal. To continue to promote green development, policymakers are advised not to separate water ecological environment protection from economic development, and need to take “green rise” as the best path for the future development of Jiangxi Province, so as to achieve mutual benefit and win-win results between water ecology and economy.

### 3.2. Estimation Results of the Degree of Coupling Coordination and Discussion

We select, 2009, 2014 and 2019 as representative years for this part of investigation, according to the actual development of new-type urbanization and water ecological environment in Jiangxi Province and its 11 cities. The state of coupling coordination between new-type urbanization and water ecological environment in various cities in Jiangxi Province is explored by using the aforementioned relevant calculation formula of the coupling coordination model. Table 4 displays the specific calculation results.

From Table 4, we can see that the overall coupling coordination degree of new-type urbanization and water ecological environment in Jiangxi Province in 2009 was 0.24, and the coupling coordination degree was in a moderate maladjustment recession state as a whole. Among the 11 cities, the coupling coordination degree of 8 cities (Nanchang, Jingdezhen, Jiujiang, Xinyu, Ganzhou, Ji’an, Yichun and Shangrao) was in a moderate maladjustment recession-low coupling coordination stage; that of Pingxiang was in a severe maladjustment recession state; that of Yingtan city was 0.06, at [0.00, 0.09], falling into the extreme maladjustment recession type, and only the coupling coordination degree of Fuzhou city belonged to the mild maladjustment recession stage in 2009. In 2014, the coupling coordination degree of the new-type urbanization and water ecological environment in Jiangxi Province was somewhat improved. The average coupling coordination degree was in the state of moderate maladjustment recession-low coupling coordination in Jiangxi Province and 11 cities. Except in Pingxiang and Yingtan that the coupling coordination degree was in the state of [0.00, 0.09], belonging to the extreme maladjustment recession type, the other 9 cities were in the stage of moderate maladjustment recession-low coupling coordination in 2014. In 2019, the coupling coordination degree of Jiangxi province belongs to the type of mild maladjustment recession. Nanchang, Xinyu, Ji’an, Fuzhou and Shangrao all belong to the stage of mild maladjustment recession-low coupling coordination; only Pingxiang and Yingtan are in the state of extreme maladjustment recession and severe maladjustment recession, respectively, with the coupling coordination degree of the former being 0.09, and that of the latter being 0.14. The coupling coordination degree is on the verge of maladjustment recession-moderate coupling coordination in Jingdezhen, Jiujiang, Ganzhou and Yichun, showing that the coupling coordination is gradually developing and optimizing in Jiangxi Province.

On the whole, in 2009, 2014 and 2019, the coupling coordination degree of new-type urbanization and water ecological environment has developed from moderate maladjustment recession to mild maladjustment recession in Jiangxi Province and its 11 cities, and improved from low coupling coordination to moderate coupling coordination. In addition, in terms of the coupling degree in 2009, 2014 and 2019, the overall coupling degree of Jiangxi Province is high, and the coupling degree in most cities is close to 1, with some cities even reaching 1. It shows that there is a strong correlation between the new-type urbanization and water ecological environment in Jiangxi Province, and the coupling effect is good, although not every city has reached a high coupling coordination and optimal coupling coordination stage, and the overall coupling coordination degree remains low. The results are similar to those of Zhu et al. [35] on the coupling coordination degree of new-type urbanization and ecological environment, but not all consistent with the results of Yang et al. [36] on the coupling coordination degree of urbanization and ecological environment. The latter found that the decoupling state of urbanization and ecological environment was mainly characterized by weak decoupling and expansion connection.

### 3.3. Regression Results and Discussion

#### 3.3.1. Regression Results

We use Eviews10.0 software (IHS Markit Ltd., Englewood, CO, USA) to estimate random effect panel Tobit regression model (10), and the results are shown in Table 5. We can see that the regression coefficients of investment in scientific and technological innovation, opening-up degree and government capacity are positive, indicating that these three factors have a positive correlation with the coupling coordination degree of new-type urbanization and water ecological environment. Hence, increasing investment in scientific and technological innovation, opening-up degree and the proportion of regional fiscal expenditure will help to promote the coupling coordination development of new-type urbanization and water ecological environment to a certain extent. In addition, the regression coefficients of economic development level and resource agglomeration ability are −4.2301 and −3.6003 respectively, which fail to pass the significance test, indicating that the two factors have no statistically significant negative effect on the coupling coordination degree. This is perhaps because at the beginning, the local government initially focused on extensive economic development, with large resource consumption and low resource utilization efficiency in the process of urbanization development in Jiangxi Province. After entering the new-type urbanization stage, the direction and orientation of economic development began to change to an intensive mode, and the resource utilization efficiency gradually improved. The education level and the industrialization level have a significant negative effect on the coupling coordination degree, indicating that the lagged development of education, and the lack of high-quality talents are not conducive to the improvement of water ecological environment in the process of promoting urbanization in Jiangxi Province. At the same time, it shows that heavy industry accounts for a large proportion of industrial development, causing certain damage to water ecological environment. Therefore, Jiangxi Province would need to change the mode of economic development, vigorously develop education, promote the upgrading of industrial structure, reasonably optimize the allocation of resources, speed up the construction of resource-saving and environment-friendly cities, and promote the further coupling coordination development of new-type urbanization and water ecological environment in Jiangxi Province cities. The regression results are similar to those of Zhao et al. [23] on the driving factors behind coupling coordination degree between new-type urbanization and ecological environment, and also similar to those of Li and Zhang [41] on the driving factors behind coupling coordination degree between new-type urbanization and ecological efficiency.

#### 3.3.2. Robustness Test

To test the robustness of the baseline empirical results, we first experiment with an alternative measure of the coupling coordination degree, that is, we re-measure the coupling coordination degree based on the improved coupling coordination degree model by Zhang et al. [42]. The regression results are shown in Table 6. Secondly, the system GMM (Generalized Method of Moments) is used to estimate the random effect panel Tobit model, where we select the independent variable lagged by two periods as the instrumental variable, to obtain reliable regression results. The regression results are shown in Table 7. Table 6 and Table 7 show that the size and significance level of the estimated coefficients of the economic development level, resource agglomeration ability, investment in scientific and technological innovation, education level, opening-up degree, government capacity and industrialization level on the coupling coordination degree between new-type urbanization and water ecological environment have remained more or less the same. Thus, this study passes the robustness test.

## 4. Conclusions

How to improve the water ecological environment level in the process of urbanization has become one of the urgent problems facing policymakers. Taking 11 cities in Jiangxi Province as the research object, this paper establishes the evaluation indicator system of new-type urbanization and water ecological environment. Using entropy method, coupling coordination degree model and random effect panel Tobit model, we explored the coupling coordination degree and its driving factors of new-type urbanization and water ecological environment in Jiangxi Province from 2009 to 2019. The research conclusions and findings are as follows:

First, the new-type urbanization level in Jiangxi Province showed a steady upward trend from 2009 to 2019. The new-type urbanization level rose from 0.049 in 2009 to 0.14 in 2019. Especially after 2016, the upward trend was more prominent. Although the new-type urbanization level of all cities experienced a certain degree of fluctuation, the overall development showed a good trend. The water ecological environment level in Jiangxi Province tends to be stable and slowly rising from 2009 to 2019. The rising range is not very large, and the comprehensive score of water ecological environment in most cities is lower than 0.1, suggesting that the water ecological environment in Jiangxi Province is not yet optimal and there is room for improvement.

Second, in 2009, 2014 and 2019, the coupling coordination degree of new-type urbanization and water ecological environment in Jiangxi Province and its 11 cities showed an upward trend, from moderate maladjustment recession to mild maladjustment recession, and from low coupling coordination to moderate coupling coordination. In addition, from the perspective of coupling degree, the coupling degree in most cities is close to 1, with some cities even reaching 1. However, not all cities have reached a high coupling coordination and optimization coupling coordination stage, and the overall coupling coordination degree remains low.

Thirdly, from the perspective of driving factors, the investment in scientific and technological innovation, opening-up degree and the government capacity are positively correlated with the coupling coordination degree of new-type urbanization and water ecological environment, while the economic development level, resource agglomeration ability, the education level and the industrialization level are negatively correlated with the coupling coordination degree.

Based on the above results and the actual development state of Jiangxi Province, this paper puts forward the following policy recommendations to further promote the coupling coordination development of new-type urbanization and water ecological environment:

First of all, Jiangxi Province and its cities can strengthen the supervision of industrial parks and mining areas, constantly improve the relevant legal systems, increase penalty on illegal sewage discharge by enterprises, and improve the awareness of water ecological environment restoration. It is necessary to further increase investment in water ecological environment protection funds, continuously establish and improve ecological environment infrastructure protection facilities, ensure that the sewage is discharged into the river after purification, and reduce damage to the water ecological environment.

Secondly, there is a need to grasp the balance between urbanization development and water ecological environment protection. The new-type urbanization and water ecological environment are interdependent, and can interact and promote each other. Hence they cannot be separated mutually in policymaking. Jiangxi Province and its cities need to focus on both economy and ecology, take both measures, firmly establish the concept of urban development in the new era, change the previous extensive (and loose) economic development model, reasonably allocate resources, so as to realize the highly coupled and coordinated development of new-type urbanization and water ecological environment.

Moreover, Jiangxi Province can increase opening-up degree and improve the open economy level, give full play to the positive role of the government, increase the proportion of regional fiscal expenditure, and stimulate market vitality. At the same time, it is suggested to continue to increase investment in scientific and technological innovation, vigorously promote the development of innovative cities, and improve urban governance, so as to realize the coordinated and healthy development of urbanization and water ecological environment.

## Figures and Tables

**Figure 1 ijerph-19-09998-f001:**
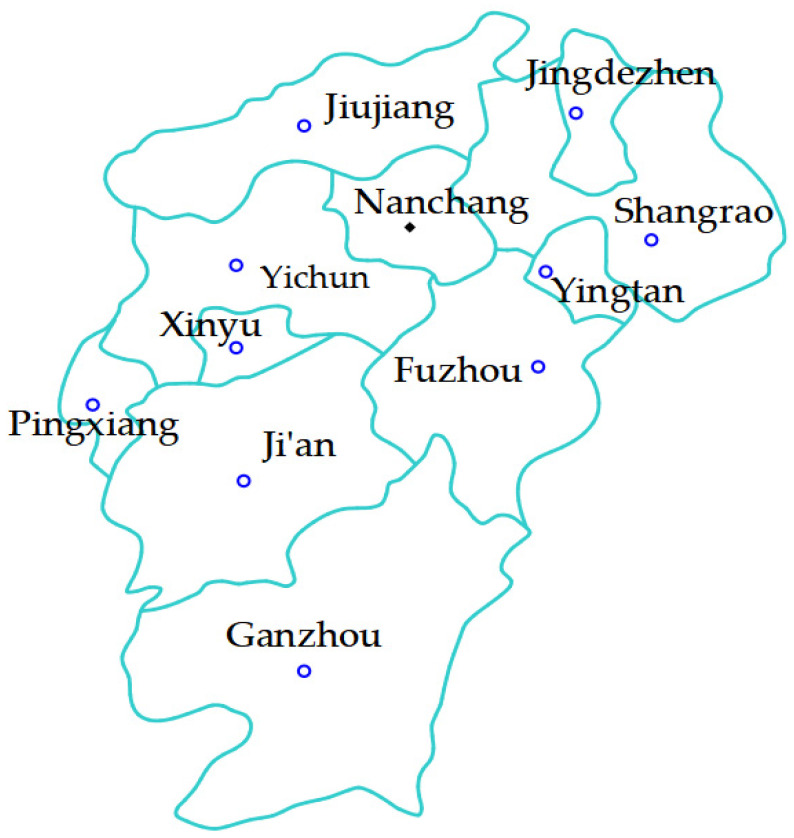
Map of Jiangxi Province.

**Figure 2 ijerph-19-09998-f002:**
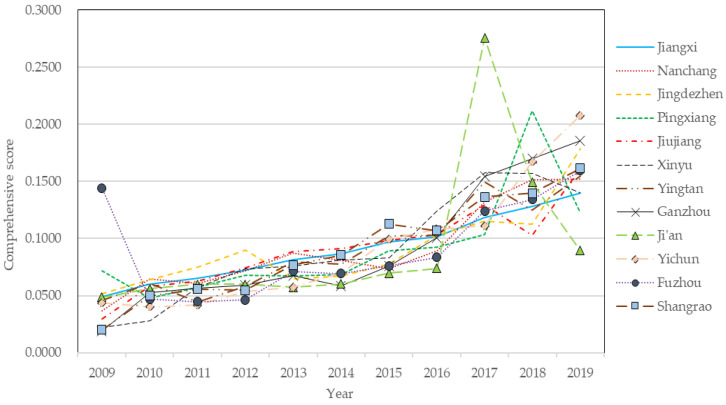
New-type urbanization level of Jiangxi Province and cities.

**Figure 3 ijerph-19-09998-f003:**
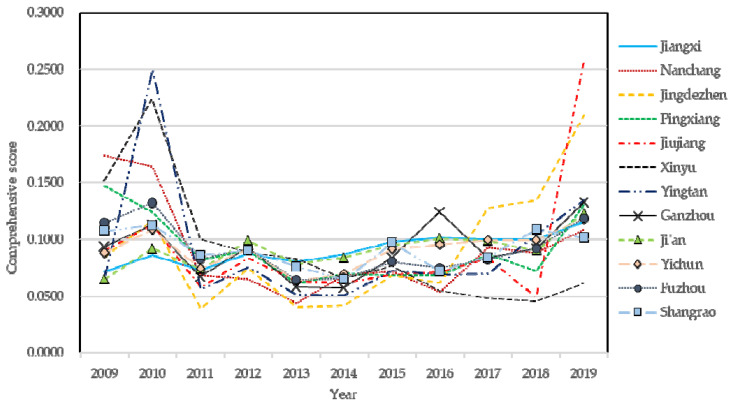
Water ecological environment level of Jiangxi Province and cities.

**Table 1 ijerph-19-09998-t001:** Indicator system and weight of coupling coordination.

System Layer	Criterion Layer	Indicator Layer	Weight	Indicator Efficacy
New-type urbanization system	Population urbanization	Number of urban registered unemployed (person) Urbanization rate of permanent residents (%) Proportion of employees in the secondary industry (%) Proportion of employees in the tertiary industry (%)	0.0091 0.0364 0.0141 0.0354	− + + +
	Economic urbanization	Per capita GDP (Yuan) Growth rate of regional GDP (%) Proportion of tertiary industry in regional GDP (%)	0.0585 0.0082 0.0308	+ + +
	Social urbanization	Total retail sales of social consumer goods (10^4^ Yuan) Number of museums (pcs) Number of internet broadband access users (10^4^ households) Number of hospital beds (pcs) Urban gas popularity rate (%) Science and technology expenditure of the whole city (10^4^ Yuan) Education expenditure of the whole city (10^4^ Yuan)	0.0820 0.0538 0.0835 0.0626 0.0020 0.1193 0.0669	+ + + + + + +
	Spatial urbanization	Proportion of urban construction land in urban area (%) Park green area (ha) Built up area (km^2^) Urban road area per capita (m^2^) Green coverage rate of built-up area (%) Green space rate of built-up area (%)	0.1068 0.0611 0.0818 0.0294 0.0282 0.0301	+ + + + + +
Water ecological environment system	Water ecological environment pressure	Sewage discharge (10^4^ m^3^) Water consumption population (10^4^ people) Domestic water consumption of urban residents (10^8^ m^3^) Total industrial water consumption (10^8^ m^3^) Urban public water consumption (10^8^ m^3^) Industrial wastewater discharge (10^4^ tons) Leakage volume of public water supply (10^4^ m^3^) Water penetration rate (%)	0.0130 0.0145 0.0333 0.0230 0.0178 0.0114 0.0115 0.0095	− − − − − − − +
	Water ecological environment status	Total water resources (10^8^ m^3^) Total water supply (10^8^ m^3^) Water resources per capita (m^3^) Water supply per capita (m^3^) Annual precipitation (10^8^ m^3^)	0.0861 0.1557 0.0426 0.1147 0.0914	+ + + + +
	Water ecological environment response	Urban sewage treatment rate (%) Total sewage treatment (10^4^ tons) Sewage treatment capacity of sewage treatment plant (10^4^ m^3^/day)	0.0061 0.1797 0.1898	+ + +

**Table 2 ijerph-19-09998-t002:** Classification of coupling coordination degree types.

Serial Number	Coupling Coordination Degree	Coupling Coordination Type	Coordination Category
1	0.00~0.09	Extreme maladjustment recession	Low coupling coordination
2	0.10~0.19	Severe maladjustment recession	
3	0.20~0.29	Moderate maladjustment recession	
4	0.30~0.39	Mild maladjustment recession	
5	0.40~0.49	On the verge of maladjustment recession	Moderate coupling coordination
6	0.50~0.59	Barely maladjustment recession	
7	0.60~0.69	Primary coupling coordination	High coupling coordination
8	0.70~0.79	Intermediate coupling coordination	
9	0.80~0.89	Good coupling coordination	Optimal coupling coordination
10	0.90~1.00	Advanced coupling coordination	

**Table 3 ijerph-19-09998-t003:** Driving factors of coupling coordination degree.

Variable Type	Variable Name	Variable Symbol	Variable Description	Unit
Dependent variable	Coupling coordination degree	*D*	Calculation results of coupling coordination degree model	—
Independent variable	Economic development level	*eco*	Per capita GDP	USD/person
	Resource agglomeration ability	*res*	Population density	Person/km^2^
	Investment in scientific and technological innovation	*tech*	Proportion of science and technology expenditure in financial expenditure	%
	Educational level	*edu*	Proportion of education expenditure in financial expenditure	%
	Opening-up degree	*open*	Actual amount of foreign capital utilized per capita	USD/person
	Government capacity	*gov*	Proportion of regional fiscal expenditure in GDP	%
	Industrialization level	*indu*	Proportion of added value of secondary industry in GDP	%

**Table 4 ijerph-19-09998-t004:** Coupling degree, coordination degree and coupling coordination degree.

Region	2009			2014			2019		
	C	T	D	C	T	D	C	T	D
Jiangxi Province	0.98	0.06	0.24	1.00	0.09	0.29	1.00	0.13	0.36
Nanchang	0.76	0.11	0.28	1.00	0.07	0.27	0.99	0.13	0.36
Jingdezhen	0.97	0.07	0.26	0.97	0.05	0.23	1.00	0.19	0.44
Pingxiang	0.94	0.11	0.10	1.00	0.07	0.07	1.00	0.13	0.09
Jiujiang	0.86	0.06	0.23	0.98	0.08	0.27	0.97	0.21	0.45
Xinyu	0.66	0.09	0.24	0.99	0.07	0.27	0.92	0.10	0.30
Yingtan	0.98	0.06	0.06	0.98	0.06	0.06	1.00	0.14	0.14
Ganzhou	0.75	0.06	0.21	1.00	0.06	0.24	0.99	0.16	0.40
Ji’an	0.99	0.06	0.24	0.99	0.07	0.27	0.99	0.11	0.32
Yichun	0.94	0.07	0.25	1.00	0.07	0.26	0.96	0.16	0.40
Fuzhou	0.99	0.13	0.36	1.00	0.07	0.26	0.99	0.14	0.37
Shangrao	0.73	0.06	0.22	0.99	0.08	0.27	0.97	0.13	0.36
Mean value	0.96	0.08	0.24	0.99	0.07	0.23	0.98	0.14	0.33

**Table 5 ijerph-19-09998-t005:** Regression results.

Variable	Coefficient	Standard Deviation	*t* Statistic	*p* Value	Adjust-R^2^	D.W	F
*cons*	0.5214	0.0654	7.9770	0.0000 ***	0.4959	0.9699	17.8655
*eco*	−4.2301	3.3803	−1.2502	0.2138			
*res*	−3.6003	3.0901	−1.1646	0.2466			
*tech*	0.0119	0.0053	2.2411	0.0270 **			
*edu*	−0.0033	0.0019	−1.7908	0.0760 *			
*open*	0.0001	6.3302	1.9792	0.0502 **			
*gov*	0.0003	0.0001	1.8214	0.0712 *			
*indu*	−0.0043	0.0009	−4.8338	0.0000 ***			

Note: *, ** and *** indicate that the variable is significant at the level of 10%, 5% and 1% respectively.

**Table 6 ijerph-19-09998-t006:** Results of robustness test (1).

Variable	Coefficient	Standard Deviation	*t* Statistic	*p* Value
*cons*	0.4678	0.0607	2.1632	0.0351 **
*eco*	−4.1263	3.3739	−1.2817	0.2034
*res*	−3.5170	3.0961	−1.2085	0.2309
*tech*	0.0192	0.0043	2.2651	0.0263 **
*edu*	−0.0037	0.0018	−1.8104	0.0821 *
*open*	0.0008	6.3273	1.9812	0.0487 **
*gov*	0.0024	0.0006	1.8338	0.0680 *
*indu*	−0.0045	0.0004	−3.6560	0.0000 ***

Note: *, ** and *** indicate that the variable is significant at the level of 10%, 5% and 1% respectively.

**Table 7 ijerph-19-09998-t007:** Results of robustness test (2).

Variable	Coefficient	*p* Value
*cons*	0.5024	0.0463 **
*eco*	−3.9007	0.1192
*res*	−3.0143	0.1248
*tech*	0.0159	0.0320 **
*edu*	−0.0032	0.0271 **
*open*	0.0045	0.0737 *
*gov*	0.0068	0.0056 ***
*indu*	−0.0037	0.0129 **
Wald test	1045.6541	
Sargan test	0.2320	
Arellano-Bond AR (1)	0.0052	
Arellano-Bond AR(2)	0.2356	

Note: *, ** and *** indicate that the variable is significant at the level of 10%, 5% and 1% respectively.

## Data Availability

Not applicable.

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
