# Peer review of "Study on the Coupling Coordination between New-Type Urbanization and Water Ecological Environment and Its Driving Factors: Evidence from Jiangxi Province, China"

_ijerph, 2022, doi:10.3390/ijerph19169998_

Round 1

Reviewer 1 Report

This paper used the entropy method, coupling coordination degree model and random effect panel Tobit model to study the coupling coordination development status between new-type urbanization and water ecological environment, and its driving factors in Jiangxi Province. The author has made some contributions to model innovation. However, there are still some main problems in this paper, the comments are as follows:

1. The structure of the manuscript must be changed. Authors put the evaluation method and results in section 3, the coupling model and results in section 4, and the driving factors model and results in section 5. That is not a typical way to structure a manuscript. Authors need to systematically explain the research models and indicators in a separate chapter, and then discuss the research results in another section.

2. In the literature review section, the author did not introduce the advantages of the model and the source of the model. The originality and novelty of the paper needs to be further improved and clarified.

3. Section 2.1 suggested add a figure attaching study area (here - 11 cities of Jiangxi Province)

4. The section 2.2 should be merged under Methods section.

5. The data source of this manuscript is missing.

6. Line 320-354, the conclusion of results needs to be improved. Have any scholars done similar research? What is the difference between your results and those in the existing literature? Authors need to add literature to support the results of this manuscript. Lines 306-406, the empirical results also need literature support.

7. Table 5, the key element of regression is missing. What is the goodness of fit check of the model? Please show the Adjust-R2 of the model. If the reader can't see it, the credibility of the regression will be in doubt.

8. The structure of the manuscript is chaotic, for example, Line 378, Table 5 seem to be a part of Section 5.2. Result Analysis.

9. In the conclusion part, the author should first summarize the research questions, models and methods. Important results and management insights should then be provided. It is suggested that the language should be concise.

10. The policy implications need to put forward according to the conclusions

11. The format of the references needs to be modified. Please check the format requirements of the citation of IJERPH carefully. The years of the references does not appear in the text. Please check carefully and modify the full text.

12. There are also some careless mistakes. Line 290 introduce formula (8), while Line 303 introduce formula (10) and formula (11), however, where is formula (9)? Additionally, there is another formula (11) in Line 369. Furthermore, Line187, the suffix of formula (1) (2) is placed on the next line, which should be placed at the right end of the formula.

13. The document contains a number of English mistakes. I would recommend having it proofread by a professional.

Reviewer 2 Report

The article represents current and important approach to urbanisation and its relation to natural ecosystem.

In the part Introduction the term 'new-type urbanisation' should be explained. Definitions given later in the section with methods description should be moved to beginning of the considerations.

In the Introduction section the main aim of the work should be mentioned. Moreover, the main stages of conducted analysis should be clearly indicated.

According to Journal requirements, methods used in the study and research results should be presented in separate sections. This would make the presented content clearer and easier to understand.

It In the paper some elements of methods are described in section 2 and the other in section 3.

Authors define new-type urbanisation as ‘an intensive, intelligent, green and low-carbon urbanization’.

How these features: intensive, intelligent, green and low-carbon are considered in four aspects constructing evaluation system. And going further, which indicators selected by Authors represent these important features of new-type urbanisation? The Author's choice of indicators should be justified taking into account the definition of new-type urbanisation and maybe also other factors.

The section 6 should be rather titled 'Conclusions'. Presented countermeasures are only very general, and they are the part of conclusions.

Round 2

Reviewer 1 Report

The authors have carefully revised the comments I raised last time. I have no other suggestions.